# Education and Heritage of Medieval Warfare. A Study on the Transmission of Knowledge by Informal Educators in Defensive Spaces

**Darío Español-Solana \* and Jesús Gerardo Franco-Calvo**

Specific Didactics Department, University of Zaragoza, 50009 Zaragoza, Spain; jgfranco@unizar.es
\* Correspondence: despanol@unizar.es

**Abstract:** Historical reenactment is becoming a top-tier teaching tool in the countries of Southern Europe. In Spain specifically, this discipline is experiencing a boom as a heritage education method, particularly in informal settings. This article is the outcome of a qualitative research study of the results obtained from one hundred and fifteen educators from historical reenactment groups. The study analyses the methods used by the exponents of this discipline to teach war in the Middle Ages, specifically in three Spanish castles dating from the 11th to the 12th centuries. It has made it possible to analyse how the educational discourses are organised in relation to Medieval war within military spaces from this period, and how historical reenactment is a coadjutant in the construction of teaching/learning spaces from a heritage education perspective.

**Keywords:** historical reenactment; conflict; heritage teaching; informal education; war spaces; warfare

## 1. Introduction

The point of departure set forth by the researchers of this study is to analyse which educational discourses are used in informal settings in relation to military history, and in particular, war in the Middle Ages. These discourses take place in heritage centres or museums, in which the military past linked to the heritage preserved and spread is explained. The 21st century and post-modernity have combined to build a story around war in the Middle Ages that relativises or even trivialises the conflicts; literature, films, television or video games are the main agents of this trend. With a view to educate for peace, this study aims to research and analyse the information given about this phenomenon through a very specific educational sector: the associative agents or groups using historical reenactment to educate in medieval war and conflict for the purposes of heritage education.

We firmly believe that hiding war and its horrors leads to nothing but ignorance, the same ignorance that is largely responsible for starting a great many of the wars. To know about the wars of the past is to promote a conscientious reflection on the same, an essential process for human growth, common sense and intellectual and behavioural maturity. Hence, we firmly believe that to show war is the best thing we can do to educate for peace [1] and thus create a more advanced, cultured society grounded in knowledge [2].

It is nonetheless true that in the current moment in time, the distance from which we approach it is equal to or greater than the distance from which we relativise it, which is why it is necessary to know that it exists and has made us the way we are. It is not so much a question of showing its horrors as it is of educating that they occurred. It is important to remember that in the learning and teaching spaces—both formal and informal—that may evolve in the scenarios that were once sites of war "a terrible true story is brought close" ("Un acercamiento a una terrible historia real") [3].

The initial hypothesis of this research is that the museum discourse applied in heritage settings directly related to medieval conflict by educators who use historical reenactment lacks a general perspective in relation to the historical period and a reflection on the same.

The phenomenon of war is isolated, hinging on specific or anecdotic aspects, such as elements of the material culture, tactical, strategic or poliorcetic data without orchestrating a discursive frame that links with the knowledge the audience has about the medieval military past to consolidate general and complex constructs implicit in the historical period, such as causes and consequences, long time structures or reflections enabling the student to think historically. This discourse has also been observed in a certain way to be tainted with clichés and popular elements which heritage education finds structurally difficult to rid itself of, despite also invalidating and annulling the same through historical reenactment, as we are aware. All of these issues constituting the exploratory hypothesis aim to draw conclusions for the subsequent development of methodological foundations to enable the correct elaboration of these discourses, in such a way that they come as accurately close as possible to an education for peace.

The objective of this research is two-fold. On the one hand, it aims to analyse the scope of the museum-centred discourse in relation to historical reenactment as a heritage education tool in the subject of war and conflict in the Middle Ages, with a view to an education for peace and the development of citizen competences. And on the other, along the same lines, it aims to understand which discursive aspects historical reenactment focuses on in its educational role in relation to the war phenomenon as a whole.

Additionally, this study also includes other secondary objectives, in turn subsumed under the overall objective. In this research we aim to:

- Gain an understanding of how the historical reenactor of the Middle Ages documents the cultural material they reconstruct and explain.
- Verify the level of scientific praxis the reenactor imprints on the documentation, and subsequently the education, process.
- Explore the reenactor's own consideration as an informal heritage educator of the work they perform.
- Analyse how the reenactor of the Middle Ages tackles the historical contents associated with war heritage.
- Analyse whether the war phenomenon is explained and divulged out of context, or through a *continuum*, linking facts and historical periods for a more complete communication, implicit in an education through historical structures rather than isolated events.
- Ascertain to what extent the educational praxis of historical reenactment contributes to the achievement of the historical and citizen education goals.
- Understand whether historical reenactment in the area of war is aided by educational resources such as empathy or interaction with the audience in informal contexts.
- Analyse how and to what extent the educational discourse on medieval conflict is belied by notions garnered from or influenced by popular culture.
- Ascertain the extent to which the historical reenactor values their praxis as a positive heritage education tool, and how it contributes to an education for peace.

## 2. Educating on Conflict: A View from Southern Europe

Tackling conflicts and the wars resulting from them may trigger unproductive and problematic ideological debates in education, which is why it is considered an unpleasant topic that tends to be avoided [4]. This is further aggravated in the event of recent civil wars as, according to Hernàndez Cardona [5], war represents violence with armed groups that wish to dominate a spatial area and the human and economic resources of the same. This is why he believes the aspects that explain why the process of teaching/learning about war proves so difficult are:

- Parts of our history have been excluded out of fear of controversy. Besolí [6] indicates that the political and ideological component present in war has a negative impact on its inclusion in the cultural offer or its use as an educational resource, to the point that it may interfere in the historic, and therefore scientific, treatment of the same.

- The economicist approaches consider wars to be anecdotic elements within what really matters.
- In the creation of nation-states, war takes centre stage, in contrast to the neighbour's reality. Wars are taught with this nationalist undertone that fosters the development of bonds in the collective "us" [7]. The contrary reaction to that war story, will show it in a militaristic light and therefore something negative that has to be avoided.
- Pacifism, in the development of a culture for peace has pointed to war as an element to be eliminated in historical practise. Wars are manifested in a living spectacle that reflects human suffering [8].

Further to these aspects, we may ask ourselves whether there is any need to explain and analyse violent conflicts or wars in either formal or informal education. The answer is a resounding yes; the aspects mentioned are not reason enough to exclude wars and conflicts from education. Conflict enables us to see other points of view, accepting diversity or cultural, religious and ideological differences, while learning to be more tolerant and to manage our thoughts and feelings [9].

We are direct heirs of battles and their outcomes, as history has been determined by the events of the past, including battles. The organisation of the space we live in at present has been historically organised by the outcomes of these battles. Hence, they are essential to understand our present. The dichotomy idealising self and conceptualising the other [10], that separates us into "goodies and baddies" or positions events between right, ours, and wrong, those carried out by the enemy, needs to be avoided. While the past may frequently have been decided on the battlefield, the study of war has been based on the analysis of the specific facts with the decisions made by its leaders from a simplistic and easily understood perspective [1]. John Keegan [11] broke away from this dynamic in an attempt to understand the development of war, but also of the people who took part in it by focusing on other types of factors. Conflicts represent far more than a series of battles, as components of a different nature, such as social or economic, have a strong presence in them. "If one is unable to regard war as a function of particular forms of social and political organization and particular stages of historical development, one will not be able to conceive of even the possibility of a world without war" [12]. Authors such as Leandro Martínez [13] explain how war affects all the aspects and bonds of a community and threatens the survival of the State. Yet at the same time, it also goes so far as to become the cause of the development of society itself:

In the face of the terrible human consequences every armed conflict brings, making it the least desirable of all the phenomena caused by mankind as a whole, the influence wars exert on societies has been, is—and most probably will unfortunately continue to be—highly varied, worsening and even annihilating many aspects and serving to develop others, occasionally with beneficial effects. It should not be forgotten that the origins of blood transfusions, painkillers or the Internet can be found in the attempts to meet different needs generated by war in the societies that gave rise to them [13] (p. 5) ("Frente a las terribles consecuencias humanas que todo conflicto bélico lleva aparejado, y que lo convierten en el más indeseable de todos los fenómenos generados por la humanidad, a nivel colectivo, el influjo que las guerras ejercen sobre las sociedades ha sido y es –y cabe pensar que, por desgracia, seguirá siendo- muy variado, deteriorando e incluso aniquilando muchos aspectos y sirviendo para desarrollar otros, en ocasiones con efectos beneficiosos. No olvidemos que las transfusiones de sangre, los calmantes o Internet tuvieron su origen en los intentos de satisfacer diferentes necesidades impuestas por la guerra a las sociedades que les dieron origen.").

From the present-day perspective, we must, in a critical way, underline the cruelty of war, but we cannot transfer this to the historical plane as we would be ignoring the mindset of the time [1]. The correlation between war, law, technology, religion or politics is so close that we cannot forget it when speaking of history as its influence is undeniable.

Manuela Fernández [14] reminds us how, for certain authors, war is politics continued in other mediums.

Although antimilitarist reactions link war to the formation of nations, it is important to remember that societies participated in and suffered conflicts prior to the existence of the current-day political foundations. For a long time, war was seen as the only solution presented by leaders in the face of conflicts, whether out of habit or because war was to the taste of society, with its fascination for adventure, danger or extreme violence [10]. This notion of rendering leaders or part of the population responsible has been criticised by García [1], who claims it is a form of criminalising the same rather than trying to understand the values and mindsets of each era, which is what should truly interest us about a war. The fact that these education for peace policies have allowed war to disappear from education is contradictory given that its presence on social media, television, films, comics or video games has grown owing to society's interest in the subject [1]. This interest is understandable given the relevance the listener finds in national histories [15], including wars, as they identify with them. This may explain why young people, in studies of what they know about their past and specifically about important events linked to world conflicts, engage with these national histories, imbuing them with meaning in a complex and critical way [16].

A change in the approach from an aesthetic dimension to a critical dimension must be considered. It is possible to promote both historical thinking and critical thinking through history, participating in an education that acts a driver of human rights. Conflicts are one more element of the past through which we can learn to decipher the present, in addition to understanding the world and ourselves better [2]. There is no need to fear including war and conflict in the classroom, or out of, it given that they work towards peace, as long as they are not approached from ideological perspectives. López Facal [17] insists that the wars that marked our history must be explained and that conflicts should not be hidden, as by explaining them we develop the capacity to defend our own points of view, while also respecting the opponent's. This fosters critical thinking, which as Hernàndez Cardona y Rojo [3] indicate, combines with the idea of making a denouncement in favour of peace through knowledge about war and its context.

These ideological perspectives must be avoided in our approach to conflicts as history, like science, should not be manipulated for partisan purposes. On the contrary, scientific knowledge must form the basis of our work, using past evidence to obtain it, with a view to developing educational proposals.

## 3. Reflections on the Teaching of Medieval War in Spanish Education

Although this research revolves around the informal educational context, it is nonetheless important to associate the start of the research with a series of epistemological reflections on how war in the Middle Ages is taught in formal Spanish education, specifically in primary and middle education. There is no doubt that the epistemological assumption of the teacher lies behind the nature of the teaching type they will implement. The scientific paradigm chosen will determine the selection and filtering of contents, the methodological resources or the design of the curricular elements needed to evaluate the teaching/learning processes [18,19]. The predominant theory in our education programme or design will form the backbone of the knowledge bases that in turn allow us to answer three fundamental questions that every formal educational fact must address: what is being taught, why is it being taught and for what purpose is it being taught [20]. In light of these reflections, it is necessary to clarify that the study of how war is taught in primary and middle education in Spain, whether obligatory or not, has been overwhelmingly conditioned by this paradigm.

If we accept that in recent years the Social Sciences teaching has partially focused on educating towards a critical citizenship [21–23] and the adoption of democratic values [24] it is also true that, based on different hypotheses and epistemological approaches, war, adorned with all its current and historical accessories, requires a very specific type of treatment to be included in the new curricular plans [1,25]. These authors, based on the

approach that supports knowledge of history through Social Sciences education, allude to "The historical perspective in the construction of democracy, the historical memory and historical awareness as a consciousness of time; that is, the relationships between the past, the present and the future", as a capital element in the construction of a youth with democratic values" [24] (p. 359) ("La perspectiva histórica en la construcción de la democracia, la memoria histórica y la conciencia histórica como conciencia temporal, es decir, las relaciones entre el pasado, el presente y el futuro", como un elemento capital en la construcción de una juventud con valores democráticos"). It is also a fact that the teaching of war tends to be omitted from these plans. In reality, it appears obvious that if war is currently filtered in among the curricular contents or even the methodological processes to educate in citizenship and in democratic values in primary and middle education it is precisely as a pretext to contribute to this education in citizenship. Thus, bringing contemporary war processes such as the Spanish Civil War or the Second World War into the classroom is justified by that interrelationship with it. Nonetheless, in this case war is used as an argumentative vessel for an educational task that, though close to it, is at the same time removed from it. It is worth noting how different this would be if we were to consider studying the 14th-century War of the Two Peters for the same purpose. Therefore, the inclusion of armed conflicts within the curricular contents to educate in these values will logically be limited to contemporary facts linked to our own current paradigm of citizenship.

Nonetheless, if we define a working space in which to position war among the contents to teach within Social Sciences, it must irrefutably be included in the History syllabus. In this context, however, it is our belief that there is a tendency to make two errors. The first lies in the fact that at present war is not taught unless a specific conflict is fundamental within a series of specific contents from a specific period. That is, war processes or conflicts are not mentioned if they are not of capital importance to understanding the historical context and, evidently, the war phenomenon is not prioritised over other paradigms of primary learning such as the economy, society, the arts or other aspects currently included in the Social Sciences and History contents of the Spanish syllabus. Related aspects are covered in the study by Lopez, Carretero & Rodriguez-Moneo [26], linked to the teaching and construction of identity-building or national discourses through the prolonged war campaign of the *Reconquista*. However, this approach must be considered erroneous insofar as it separates war from other historical processes that are not understood without it, approaching the study of history through sealed and isolated compartments. In fact, the war-related phenomena are behind a large part of the historical changes and the explanation of the same, yet when it comes to teaching history they are relegated as accessory elements in the belief that it would be better if they hadn't existed, when, paradoxically, even if it weighs on us, they have made us who we are.

The second error is the direct denial or elimination of the conflict from any educational programme. Behind this debate lies the false belief that to educate for peace it is essential not to explain what war is [27]. As Capmany, González y Marín state:

> If we are incapable of breaking away from these ideas we will be collaborating in the justification of imperialism. The educational discourse built around war as an axis of history becomes one more tool in the service of power. Thus, we will be strengthening the national arrogance that justifies the dominion or exclusion of some over others. We will, in short, be blessing the flags of those who lay the ground for a war: a mass killing [28] (p. 22) ("Si no somos capaces de romper con estas claves estaremos colaborando en la justificación del imperialismo. El discurso pedagógico construido alrededor de la guerra como eje de la historia se convierte en un instrumento más al servicio del poder. Así, estaremos reforzando la soberbia nacional que justifica el dominio o la exclusión de los unos sobre los otros. Estaremos, en definitiva, bendiciendo las banderas de los que preparan el camino a lo que es una guerra: un asesinato en masa.")

Many education professionals forget that to educate is not to incite. Television, video games or certain inappropriate online contents may show outrageous scenes of violence, though that is a subject for a different debate, yet if we deprive students of the only weapon that will enable them to discover and learn reflectively what violence was and what it meant, we will be paving the way for these extracurricular stimuli.

All these issues arise as a by-product of an epistemological approach in which didactic mediation makes the mistake of not detaching itself, of previously judging the story to prevent the students from doing so themselves. It is, in that moment, tarnishing the educational fact with the teacher's personal beliefs or values [29]. It is our opinion that to teach is to foster the development of tools for reflection, critical thinking and knowledge construction, not to insert previously digested learnings. In the majority of cases, war is hidden due to a series of criteria that fall within the second and, in our view mistaken, option.

War entails too many aspects interwoven into human nature itself, however dull that may sound. It combines a violent dimension, replete with anti-values, with a social, economic, technological or thinking-related, and therefore humanistic, dimension. The study of wars is an irreplaceable tool for the development of critical thinking and reflection on conflict-solving. In fact, as some authors have indicated, to educate in conflict implies positioning it head-on. Cascón Soriano separated the "preventive" actions of this school context into three phrases: "An appropriate explanation of the conflict, including its human dimension. A knowledge of the structural changes necessary to eliminate its causes. And a promotion of conditions that create the right climate and favour cooperative relations that decrease the risk of new outbreaks, learning to tackle and resolve the contradictions before they become antagonisms" [30] (p. 14). ("Una explicación adecuada del conflicto, incluyendo su dimensión humana. Un conocimiento de los cambios estructurales necesarios para eliminar sus causas. Y una promoción de condiciones que creen un clima adecuado y favorezcan unas relaciones cooperativas que disminuya el riesgo de nuevos estallidos, aprendiendo a tratar y solucionar las contradicciones antes de que lleguen a convertirse en antagonismos"). The use of dynamics to analyse the causes of wars may give rise to spaces for reflection and debate that strengthen negotiation capacity, empathy or the establishment of convention models for the resolution of an imminent conflict; for thinking, from a more general perspective, historically [31,32]. Educational resources which are, therefore, educating for peace ("[ ... ] peace is more than just the absence of war." [33] (p. 3)).

The complexity of the military, social, economic or institutional prolegomena of a conflict can serve to develop skills like no other case of historical models offers. Hence, at the end of the day war constitutes the consummation of human violence and, therefore, can help us educate to prevent it [34]. It is not a question of showing the horrors of war, but of guiding a teaching/learning process based on past war phenomena that enables the construction of a framework for student reflection on the scope of human suffering, with a view to preventing violence in general. In this regard, the social repercussion wars have on the population is particularly interesting. According to Moreno-Vera, raising the visibility of people who have suffered war is very positive for students, allowing them to reflect on the consequences for the civil population. It gives rise to subjects such as shortages, prices or even the conceptualisation of the woman [35].

On the other hand, a lack of knowledge about military history from a holistic perspective tends to, perhaps deliberately, repudiate the fact that it houses a heterogeneous array of dimensions, not only relating to violence, reprehensible human conduct or suffering. The convergence of the violent encounters of all armed conflicts have contributed some of the most complex, lucid and brilliant passages to the history of human thinking. Military genius has served to develop profound strategic reflections to confront problems of military inferiority, the prolongation of ceasefires, the obtainment of economic resources, the growth of logistics foundations, adverse orography, negotiation, geopolitics, and an endless list of considerations innate to the field of military strategy that may serve as models to develop the strategic thinking of students. Today, the world's main military academies use these mechanisms to develop the best strategic thinking of future commands,

giving rise to real processes in which the *cognoscenti* must tackle challenges that allow them to reach satisfactory solutions, in the majority of cases on historical hypotheses that actually happened. These mechanisms, adapted for the purposes of education, constitute methodological tools of the highest level to develop fundamental student competences in the different educational stages. Traditionally, strategic thinking has erroneously been linked to competitiveness, when in reality it consists of a series of universal guidelines that are essential for the development of critical thinking, social relations or the preparation of any citizen to form part of our current societies [36].

Lastly, within these reflections on the teaching of war, we should not forget that the conflicts of the past represent a large part of the cogs that make up identities:

> [ . . . ] the teaching of history and the creation of (mainly national) identities have been entirely interconnected and largely continue to be. The genesis of this relationship between history and identity lies in the birth of the liberal State and the rise of the nineteenth-century nationalisms. In fact, in practically all western countries the generalisation of history teaching occurred from the first third of the 19th century, when the liberal States and the nationalist programmes started to impose in their educational programmes the teaching of a subject which, from that moment on, would have little to do with the humanist and citizen values it had held in the 18th century [37] (p. 334). ("[ . . . ] la enseñanza de la historia y la creación de identidades (principalmente nacionales) han estado totalmente ligadas, y en buena parte siguen estándolo. La génesis de esa relación entre historia e identidad se encuentra en el surgimiento del Estado liberal y el auge de los nacionalismos decimonónicos. En efecto, en la práctica totalidad de los países occidentales la generalización de la enseñanza de la historia surge a partir del primer tercio del siglo XIX, cuando los Estados liberales y los movimientos nacionalistas imponen en sus programas educativos la enseñanza de una materia que, a partir de ese momento, poco va a tener de los valores humanistas y de ciudadanía que había tenido en el siglo XVIII.")

It is true that post-modernity has given rise to a balance in the configuration of these, which have gone, or are in the process of going, from being fundamentally national and concomitant with the cultural and ideological precepts of the nation-states, to being considered multiple by certain authors [38]. This process is paving the way for the atomisation of cultural identities in which Social Sciences education plays a very important role [39]. Thus, in the post-modern emergence of territorial identities history tends to constitute a basic and frequent cog, particularly when the national identity models of the 20th century have been splitting into similar paradigms in line with post-modernity. Armed conflicts such as those that took place in Hastings (1066), Las Navas de Tolosa (1212), Aljubarrota (1395) or the Siege of Barcelona (1714) are perfect examples of this identity-building model in themselves, having gained protagonism decades ago in the strategies designed by the powers that be to include public policies relating to this configuration. Thus, whether we like it or not, war continues to form part of the construction of identities. And not only in a school setting, obviously.

The truly fascinating part of studying the cognitive and psychological principles involved in the didactics of history and heritage is that some of the hypotheses in existence since Comenius announced them in the 17th century have been expanded on or consolidated through the psychology of learning in the subsequent centuries, and currently through what we know about educational neuroscience. We now know that, in the different brain development stages from birth to adulthood, including childhood, puberty or adolescence, the brain construction processes experience the opening of what is known as "plastic windows" [40] (p. 40) or "critical periods" [41,42] (p. 20), in which the brain, through exogenous stimuli, builds the mechanisms of the complex cognitive network, including speech, critical thinking, calculation, among many more. Moreover, these windows can in turn be subdivided into "sub-windows" and even "micro windows", meaning the brain is predisposed to construct neuronal pathways and therefore learning in very specific,

and even time-limited, moments more so than others [40]. All of these matters will be essential in the future to train teachers, and to help build methodological structures that combine the psychology of learning with neuroscience.

Going back to the reflections on the actual teaching of war in the classrooms, of all these theories listed so far, a general pattern of justifying the use of principles that create knowledge through the war-related facts of the past and all of their associated dimensions, emerges. The verb used: justify, is not chosen by chance. It would appear that inertia, in spite of ourselves, obliges us to defend epistemological bases rather than enunciating them. This occurs as a consequence of living in a society like the Spanish which is fiercely antimilitaristic. The repudiation of all things war-related in the past tends to hide it, distance it from the present, and also from the school settings, when in reality, as we have said above, it is knowledge of the same that is going to allow us to face up to the future with guarantees. However, there is no doubt that a large part of society abhors war as a whole, without understanding that there are also learnings to be gained from the hard and bitter moments of the past.

In our opinion, the primary factor that triggers a study like this one on how to teach history in the military heritage, is the belief that war is indissoluble as a coadjutant and necessary phenomenon in the comprehension of the long-lasting structures or times. Paradigms such as the heavy feudal cavalry that, like a millpond, absorbed through its origin and military nature all the cracks in the social or economic structures of the High Middle Ages; or gunpowder, that was similarly responsible for bringing about an unprecedented change in structure and mindset that encompassed diverse contexts, are very patent general examples. Theory that works along very similar lines to what Prat puts forward in relation to the historical concepts in the teaching of history: "They have an *intensive dimension*, in so far as they describe a reality in all of its depth, an *extensive dimension* in so far as, with variations, they offer characterisations that are constants in the historical processes, a *time dimension,* in so far as they vary according to the historical period in which they occur and, finally, a *relational dimension,* in so far as they are only explicable in relation to other realities" [43] (p. 46). ("Tienen una dimensión intensiva, en la medida que describen una realidad en toda su profundidad, una dimensión extensiva en cuanto que, con variantes, ofrecen caracterizaciones que son constantes en los procesos históricos, una dimensión temporal, en la medida que varían en función del tiempo histórico en el que se dan y, por último, una dimensión relacional, en la medida que sólo se explican con relación a otras realidades"). The second lies in the pedagogical nature of the so-called centres *of interest.* Medieval war acquires practically hagiographic connotations in the popular mindset, reducing many of its complex elements to images or mental paradigms close to modern societies thanks to literature, film or television. This imaginary is riddled with *includer* links that enable us to provide a methodological frame affecting not only the war phenomenon itself, but also the general historical circumstance behind it.

It was Braudel who put forward the differentiation between short term, medium term or the *conjunctural* and long term or *structure* as fundamental elements of *Historical Time.*

> "[ . . . ] this inquiry is inevitably destined to end in the determination of social conjunctures (and even structures); and nothing can guarantee in advance that this conjuncture will have the same speed or slowness as the economic" [44] (p. 70). ("[ . . . ] esta encuesta está abocada forzosamente a culminar en la determinación de coyunturas (y hasta de estructuras) sociales; y nada nos asegura de antemano que esta coyuntura haya de tener la misma velocidad o la misma lentitud que la económica").

The long term would be a lasting construct in time. In general, it is not consciously present among us and serves as a common thread for economic or psychological paradigms involved in the historical cycles. It may go beyond time periods and conventions and transversally explains the changes and permanence of the human being in the past:

The second and far more useful key consists in the word structure. For good or ill, this word dominates the problems of the *longue durée.* By structure, observers of social questions mean an organization, a coherent and fairly fixed series of relationships between realities and social masses. For us historians, a structure is of course a construct, an architecture, but over and above that it is a reality which time uses and abuses over long periods [44] (p. 71) ("La segunda, mucho más útil, es la palabra estructura. Buena o mala, es ella la que domina los problemas de larga duración. Los observadores de lo social entienden por estructura una organización, una coherencia, unas relaciones suficientemente fijas entre realidades y masas sociales. Para nosotros, los historiadores, una estructura es indudablemente un ensamblaje, una arquitectura; pero, más aún, una realidad que el tiempo tarda enormemente en desgastar y en transformar.")

This premise connects with a concept of the future of humanity rooted in historical processes framed between transformations, that *per se* explain historical evolution [45]. Hence, the engagement of mindsets in these processes of change, far from constituting futile determinants, are replete with common threads that form the backbone of such evolutions.

It is therefore irrefutable that an exercise in abstraction in accordance with this theory may attribute the war processes a capacity that few other paradigms can offer. We may even select more or less general, or more or less trivial elements as links or points of departure through which to teach the changes in the history of Humanity. A battle, a highly strategic defensive space or a military object—the spur, the stirrup, the sword—may represent the genesis of the reflection processes needed to understand the long-term historical constructs [46]. Through these, more complex paradigms may be introduced to explain the changes that have occurred in economic, social or cultural *conjunctures* or *structures.* The interconnection of war with these elements is a highly valuable methodological tool.

This approach associated with war paradigms—or rather the humanistic dimension around war—to contextualise more complex or longer-lasting historical processes finds a very powerful ally in the theory of the *centres of interest.* This is a method coined by Ovide Decroly, midway between the 19th and the 20th centuries, the application of which includes the consideration that the student perceives the elements of reality through the *principle of globalization,* which tends to condense information or interpret what they are seeing as a whole before focusing on its parts or details [47]. Santacana and Llonch worked on this epistemological basis from the perspective of Social Sciences education for the development of methodologies under the didactics of the object [48,49].

## 4. War in the Middle Ages: Informal Education and Historical Reenactment

A contextualisation of the conceptual framework on the didactics of conflict and the didactics of war, in addition to the state of the Spanish question in: Español [50]. They mainly include defensive spaces, but also others such as battle fields or war scenes (camps, prisons, destroyed towns or villages, hospitals ... ), selected on the intuition of those involved as "sensitive, vulnerable or geostrategic" points [5]. Battle fields have become a controversial resource for the teaching of history and landscape in countries such as the United States of America, Great Britain, France or Belgium [51]. These landscapes hold evocative power as spaces for memory and oblivion, as points of reference for identity that constitute wounds or celebrations within a community [52]. Which is why the search for these scenarios is becoming increasingly common, even the old roads and entrance ways [53].

The educational possibilities of defensive spaces in which the biggest advances and learnings of each era were applied in their construction and improvement, using all the resources available for this purpose, are of enormous interest to us. For Cuenca [54] the castles and forts are among the most evocative for society when it comes to monumental heritage. The grandeur and power they communicate render them remarkable resources for the learning of history and, in particular, for the comprehension of social conflicts.

MacManus [55] points out the need to visit battle grounds. By travelling those places related to conflicts the visitor is able to imbue them with a material substance and a personal view, in turn gaining a better understanding of the past. The first-person experience makes it possible to delve deeper into the concepts, acquire the procedures and work the values. Of great interest is the proposal put forward by Cuenca [54], in relation to the Andalusian fortifications, where he positions the castles as centres of interest in the educational processes. These educational processes make it possible to cover the concepts related to said fortifications, that cannot be understood without these defensive or geostrategic processes innate to the conflicts [1]. They also allow for procedures such as the handling of historical information sources, fieldwork, spatial location or spatial measurement to be worked on. Finally, by working on values we also include all those attitudes such as education for peace, coexistence, respect for different cultures or the preservation of heritage [56], mentioned above and also reflected by Jaén [57]. This experience places the student at the centre of the learning process, as explained by Sáenz del Castillo [58], bringing them close to the historical aspects of everyday life that are omitted from the great narratives. Raising the value of these cultural landscapes may generate a very valuable learning of history and the territory, while also attracting economic resources in a historical-cultural tourism context [51].

The experiential analysis of scenarios and protagonists awakens emotions, values [58–60] and positionings of empathy that lead to increased learning motivation. This historical perspective is essential when it comes to promoting comprehension of the story, putting oneself in the shoes of a character from the past to understand their attitudes and motivations [61,62]. The necessity of an emotional engagement within the exercises of historical empathy is subject to debate [63], between those claiming the emotional component leads to a better understanding [1,61], and those who attribute it a more cognitive component, given that this emotional engagement does not constitute a form of Historical Thinking [35,62].

When analysing these heritage remains, archaeology is essential and becomes a useful tool that, as Santacana [60] points out, enables us to understand our existence, with a common, shared past that we descend from. It fosters active learning using investigation and discovery as systems that awaken motivation and curiosity [64]. Archaeology is furthermore attractive for the viewer, regardless of age, developing imagination capacity while also facilitating the introduction of historical thinking [60]. Seixas y Morton emphasise the importance of presenting this historical thinking to students to help them learn to manage the past. Furthermore, "learning to think critically about the mistakes and horrors of the past contributes to development of student's historical consciousness" [65] (p. 171).

While traditionally archaeology has been associated with the oldest times in our history, in reality it is a science that is perfectly useful for more recent times [59]. For the subject at hand, there is a subfield of archaeology known as conflict archaeology, that investigates the heritage remains linked to violent conflicts over the course of history. Hernández Cardona [66] points out that this archaeology focuses on wars, battles and fortifications, and on those collateral aspects that may prove interesting and revealing [67,68] in sites such as shelters, destroyed zones, graves [69], areas of repression . . . Conflict archaeology is a response to the deterioration of the heritage associated with war-related events, with the aim of documenting, interpreting, preserving and spreading them [58].

While a great deal remains to be defined and polished in archaeological research, very significant advances have been made with the development of very specific techniques and methods to recover and record evidence of conflict and interpret how the battles occurred [70]. Yet it is important to bear in mind that, as Hernàndez Cardona y Rojo [3] explain, the material remains generated may be very diverse and come from different conflicts, they may even not be proportional to the importance of the same. Quesada [71] warns of the risk of impoverishing the scientific discourse and research by compartmentalising according to specific interests. In spite of these warnings, advances have been made in these subfields, allowing them to gain ever-increasing protagonism in the 21st century, concentrating on modern or contemporary periods. This proximity has led to

a very significant political component affecting it which, according to authors such as Hernàndez Cardona [66] is inappropriate and may detract historical rigour. The celebration of battlefield and conflict congresses in countries like Scotland, Sweden, the United States or England, since the year 2000, has allowed this speciality to advance even further, with the use of new analytical techniques and *geographic information systems* (GIS) in data collection. Studies such as those conducted by Spennemann demonstrate the possibilities resulting from the use of new technologies applied to the study of old battle fields, in this case the Second World War [72]. At the same time, it has grown chronologically to encompass any other history period, including prehistory, to study the development of battles [70,73]. The explanation of this type of heritage remains can pose a veritable challenge for the educator, both in formal and informal education, but to manage it, as Feliú [59] states, means being able to form a historical consciousness that enables participation in society and collective decision-making in the community.

Archaeology, however, has not only permitted the study of these memory spaces from the perspective of immovable property elements but also the analysis of the movable property heritage, with objects and artefacts that speak of the conflict itself, and of the people who experienced it. Santacana y Llonch [49] demonstrate how the object has very extensive educational possibilities, establishing a series of benefits resulting from them being primary and secondary sources, from being real and tangible, for fostering imagination and being *inclusors of the mind*, while also being motivational. To understand the development of a conflict, the utensils and participants of the same need to be considered, as the weapons, the weapons of siege [74,75], the saddles and even the technology applied to the foodstuffs will be decisive for its outcome [1]. The detailed study of aspects such as weaponry makes it possible to understand the possible adopted strategy and the role and lifestyle of the individual in war [70]. In the event of more modern conflicts this weaponry may still be preserved *in situ*, with the problems the presence of major complexes or weaponry from countries in geographic locations that do not fall under their jurisdiction can cause for the conservation of the same. Management of this heritage becomes difficult, even more so if it is fruit of treaties resulting from conflicts solved via negotiation, which is why supranational organisations should be responsible for it [76]. This information enhances and completes the rest of the historical sources, such as documentary records or oral sources.

How can we be capable of joining the defensive spaces, the real or reproduced archaeological objects that appear in them, the data provided by conflict archaeology and put together an educational proposal of the highest standard?

At present, cultural habits of leisure and consumption have given rise to different ways of investing our free time, creating service industries in which, in the majority of cases, the public administrations have taken on a significant guiding and promotional role. This is the case of the culture industry of historical festivals and evocations, consolidated in the last two decades as models of local identity and tradition, which in the case of the Southern European countries has acquired peculiarities of its own [77].

In the last two decades, the Iberian Peninsula has experienced a remarkable increase in the proliferation of these events; the so-called historical festivals. In Spain, the media and public bodies also call them historical reenactments, unlike other European countries which dissociate the two concepts from an institutional and scientific perspective [77–80]. Historical festivals, though present in the majority of European countries, are a very typical product of the Mediterranean and Southern European states. Ultimately, they are a way of interpreting the phenomenon of commemorating the past through an autochthonous lens. Festivals and fun are innate signs of identity of the Mediterranean culture, which is why it is not surprising that even the distinctions and titles given to these events by various state levels is that of *"Fiesta de Interés"* (Festival of Interest). The street emerges as the place for meeting and fun, taking advantage of the past and identity to generate social and cultural projects with citizen-determined organisational criteria, with different degrees of specialisation.

Such events are structured according to the organisational criteria of the local authorities and volunteers. Their goals tend to be to boost local and territorial commerce and tourism. This model contributes to the creation of groups of local volunteers who act under the structure of tourist or cultural associations and private events management and promotion entities [81].

On the other hand, it's important to differentiate what is internationally known as historical reenactment. In Spain, though still in the early stages of growth, it is increasingly conducted on the basis of criteria grounded almost entirely in philanthropy. We understand that historical reenactment is "the practise of reconstructing uses, customs, material culture and aspects of the past in accordance with strictly scientific guidelines, to attain objectives of cultural dissemination and education" [82] (p. 335). ("La práctica de reconstruir usos, costumbres, cultura material y aspectos del pasado a partir de pautas taxativamente científicas, para lograr objetivos relacionados con la divulgación cultural y la educación"). Nonetheless, it is important to understand that for a while now the investigative dimension of this discipline has been gaining ground [74,83,84], although it does allow the inclusion of the experimental method within the historical method, the scope of which tends to be somewhat limited by the finitude of sources. Historical reenactment is therefore different to other manifestations related to history that lack this scientific framework and the irrefutable dissemination factor [78,85–87].

This discipline has proved itself to be a method capable of placing tools for reflection on the past at the disposal of different audiences [86]. This premise suggests that the study and preservation of heritage is insufficient; without the dissemination and democratisation of historical knowledge, a large part of its cultural and social value is lost [88]. Additionally, it advocates an irrefutable social function: it not only consists of disseminating and interpreting history, but the heritage it has bequeathed to us must be preserved and handed down in the future as part of our identity [80,89–92].

Historical reenactment in the south of Europe and specifically the countries of the Iberian Peninsula is undergoing a slow but progressive transformation towards academicism in the reconstruction of cultural material. Nonetheless, in spite of this progress, it is important to consider that in many aspects it tends to persist in its omittance of fundamental factors that are inherent to it beyond the capital element consistent of applying a historical method for said reconstruction. These factors are those relating to its educational activity, in both the formal and informal setting. Our point of departure, as researchers, is the consideration that the groups responsible for the reenactment which work with governmental bodies and other managers of educational or heritage centres show knowledge of that which they are recreating, but tend to lack the methodological resources relating to heritage education. This results in a number of consequences, like disconnection for example, from the perspective of formal education, the curricular contents relating to cultural material and what is shown in the different periods recreated.

When in museums or heritage sites knowledge is socialised or there is education in conflicts—particularly those of the Middle Ages, the subject at hand—through historical reenactment, our initial experience tells us that the segmentation and isolation of the contents relating to armed conflicts result in a discordant discourse, often taking it out of the context of the historical period in which it occurred. Similarly, the clichés and influences of popular culture taint the discourse with truisms and trite concepts, which at the same time blur the reality of the conflict and the medieval war, tending towards trivialisation or a normalisation that impels a lack of reflection [79].

In any case, one of the general challenges faced by these heritage education and historical reenactment projects in the coming years is to develop permeability between the academic, heritage and museum contexts. This connection will allow the discipline to mutate towards higher quality projects. Because the attainment of higher degrees of specialisation in historical rigour, heritage discourses or complex scientific subjects does not mean that the public is incapable of accessing them; in reality, the specialisation of the educators and exponents behind this will be capable of converting this complexity

into accessible, meticulous, educational and digestible programmes for all types of public without losing a serious cultural motivation, worthy of an advanced society.

Nonetheles, it is important to remember that reenactment in Spain is still far from a direct relationship between transference, the academic world and a collegial educational effort alongside the government bodies. As we have mentioned above, the vast majority of the reenactors see the discipline as a hobby. Its exponents do not yet form part of official education or museum projects for which they would receive grants or official commissions. The consequence of this is that in the majority of cases, their educational activities—if there are any—do not tend to be subject to professional requirements or regulated demands. Similarly, the joint work done between university departments or research projects with these reenactment groups remains insufficient, although it has increased slightly in recent years.

The relationship with Spanish museums is also still minority. Nonetheless, we are starting to see some groups increasingly collaborate with museum-related institutions. It is also true that, for the subject at hand, reenactment of the Middle Ages in its role of providing educational support to museums lags far behind that of other eras, such as Classical Antiquity or the Napoleonic Wars [79]. In any case, these groups rarely offer educational projects of their own when collaborating with museums, or adaptations of contents to fit the audience characteristics, or any other methodological elements innate to heritage education. These are new challenges the discipline must rise to in the future. At present, all of these issues have been exacerbated by the COVID-19 pandemic which has led to the inactivity of the majority of the groups involved in medieval reenactment, or to smaller but minority activities online. In general, the philosophy ruling reenactment activity in Spain during the pandemic has been to wait for better times.

## 5. Materials and Methods

The analysis of informal education in the area of medieval war through historical reenactment in heritage sites has been conducted as a study based on fieldwork focusing on a specific group, in its role as heritage educator: the historical reenactors. It is, therefore, an evaluation of the didactic method applied by these educators, either individually or following group action processes, in the educational usage of the heritage representation spaces, specifically those linked to the military past in the Middle Ages, following, in this case, the lines of research in Didactics of the Social Sciences put forward by Prats [93].

The methodology has consisted of an anonymous survey of different members of reenactments groups and associations, to understand how the medieval war discourse is built in the different heritage settings where they act, which socially-acquired disruptive elements are present in these discourses, and how this informal education contributes to a significant learning, education for peace or the development of historical competences in the area of citizen education. The survey was designed in Google Forms with five different sections covering: the general information; the perspective of the reenactor; the contribution of reenactment to history and heritage; the teaching of history through historical reenactment; and the teaching of war. Respondents were informed that participation was voluntary and implied consent for responses to be used to research the value of historical reenactment in teaching history and specifically teaching medieval war. Likewise, they were also informed that the data was anonymous and would be erased from the web on conclusion of the research. The survey was distributed via the social media of the actual reenactment groups that collaborated to ensure they reached the maximum number of people possible, establishing a time limit for the acceptance of responses.

Specifically, the reenactmentist educators who participated in the study are those who partake in the heritage education actions conducted yearly in three of the most noteworthy castles in the region of Aragon (Spain): the castle of Monzón (Huesca), the castle of Peracense (Teruel) and the castle of Loarre (Huesca). Thus, although the reenactors may form part of other educational projects pertaining to other specific eras, this study focused on those operating in poliorcetic heritage sites from between the 11th and the 14th centuries.



The study model through an anonymous questionnaire has been based on the implementation of a qualitative information collection tool in relation to the research goals established. Processing of the information obtained was organised according to a database with qualitative categories closely and directly relating to the research questions. The information was generated using graphs and tables for a structural analysis of the research itself, on the one hand, and as the development and publication of the results of the same, on the other. Hence, its analysis has inevitably been conducted in direct association with the objectives and the considerations prior to the study.

A series of imporant prerogatives were taken into account in the design of both questionnaire and study. The first being that it has been appraised and validated by a committee of experts in the subject. The research and questionnaire have been positively reviewed and evaluated by a committee of researchers linked to the research group the main researchers belong to, and the university institute this group is located in. The group and its team have extensive research experience in informal education, museum studies and educommunication; the comments of this appraisal have been included in the final modifications and result of the questionnaire. Additionally, though not an element of clinical research, this questionnaire has been designed in accordance with the criteria, general principles and requirements of the Declaration of Helsinki on the ethical principles for medical research involving human beings.

## 6. Results

One hundred fifteen people collaborated in the questionnaire with a large majority of males (82 of them), all of adult age, the largest age group being that of 26 to 45 years-old (50 of the respondents). Although 57% have been carrying out historical reenactment for over 5 years and participating in over 3 events per year, except this last year due to the pandemic situation, the majority do not have a work relationship with historical reenactment, or even with heritage. In spite of this, over 58% adopt great professionalism, trying to live like someone from the era, some even replicating the practises, attitudes and linguistic usages of the period being recreated.

The first data yielded by the study worthy of consideration informs us that the relationship between the exponents and the discipline is philanthropic, not professional. This may contribute to an understanding of some of the aspects linked to the educational approach used for military heritage and the war-related past in these teaching/learning settings. In any case, and in spite of this, the respondents indicate that they take a professional approach to the praxis in relation to the documentation of the cultural material and the real objectives of the educational intention that historical reenactment promotes. This aspect, however, affects the answers relating to educational planning and management of the educational contexts and situations generated during their practise. In the reenactments that questionnaire respondents participate in, both the use of the theatre and the performance of the characters, as well as the presence of a *cicerone* to explain the scenes recreated are prioritised almost equally, though the latter holds a slight advantage (Figure 1). Nonetheless, when they decide the importance of both options for a good reenactment, 72.2% consider the figure of the mediator to be recommendable, while the presence of closed scripts is more controversial, with the maximum number of opinions taking a neutral stance and leaning slightly more towards the opinion of considering them unnecessary (Figure 2). Thus, the reenactors tend to consider the use of an educational mediator a necessary resource in the educational process, acknowledging the benefits of the theatrical performance, but also considering the one-way nature of their discourse. 80% considers interaction between the reenactor and the audience fundamental, while 87% allows the public to use, experiment with and handle the reenactment material, specifically that relating to war, under their supervision. When asked about the public reenactment model that tends to be used in the events participated in, with regards to that contact with the public, 39.1% subscribe and adapt to the way in which the organisers have arranged the museum discourse and the contact with the audience. 27% recreate their scenes and offer

individualised explanations, either when the audience requires information or on the initiative of the reenactors themselves. Those who wait for the audience to congregate around their scene or set to develop explanations, workshops or a specific museum discourse, constitute 13%. Finally, 20.9% do the same, but having previously assessed the knowledge level of their audience about what they are going to explain (Figure 3). Thus, the possible professional gaps in the design and implementation of museum discourses are observed when the reenactors tend to mould themselves to fit the organisational plan of the heritage site they are operating in, adapting more to its museum or exhibition criteria, in spite of the fact that when it comes to their own praxis, the majority prepare and sequence the museum discourse of what they are going to show or explain -60.7%-.

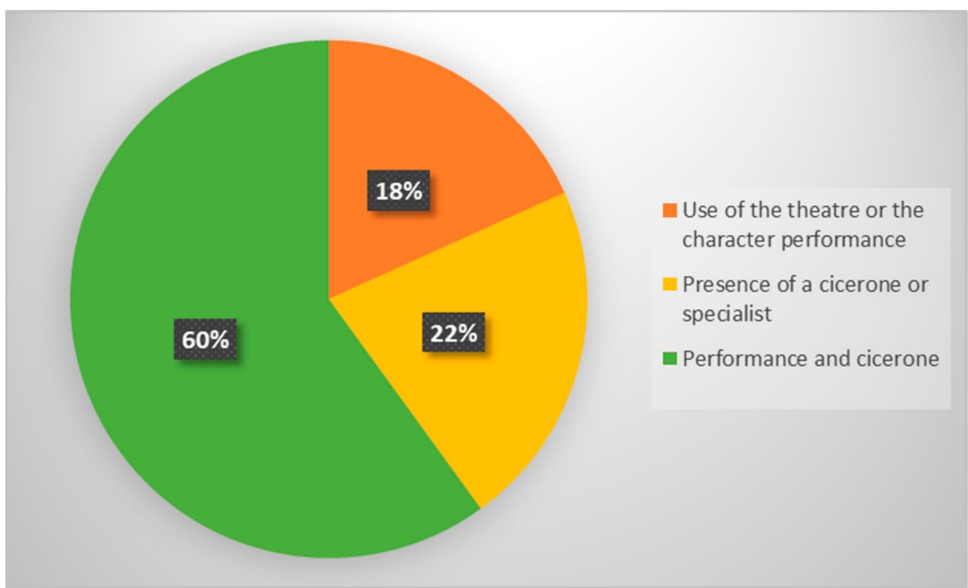

**Figure 1.** Fundamental method in knowledge transmission.

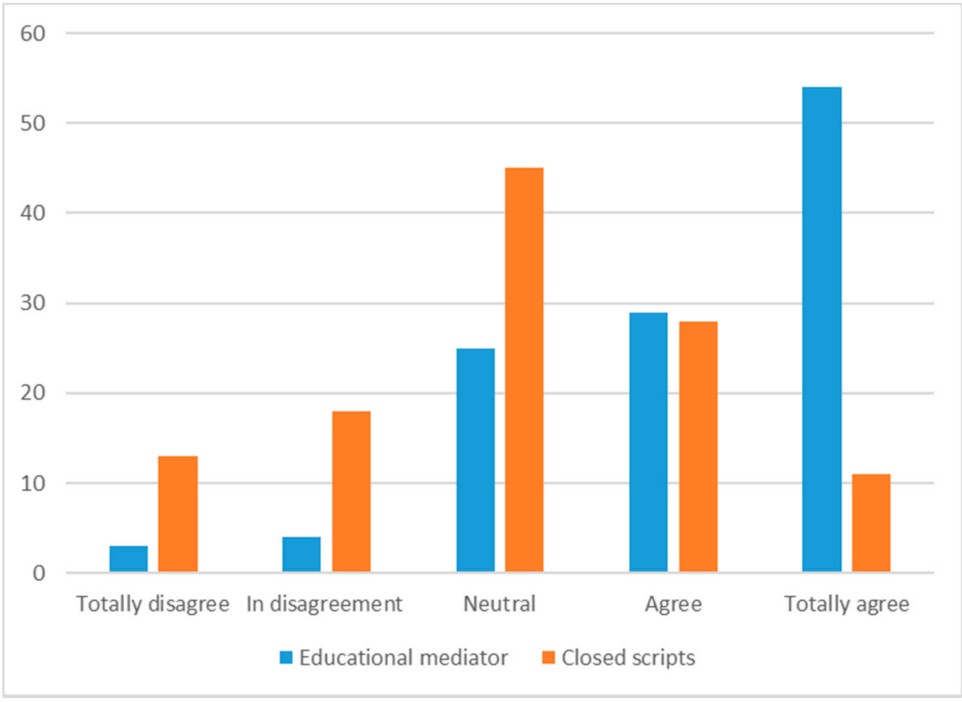

**Figure 2.** Mediation between the reenactment and audience.

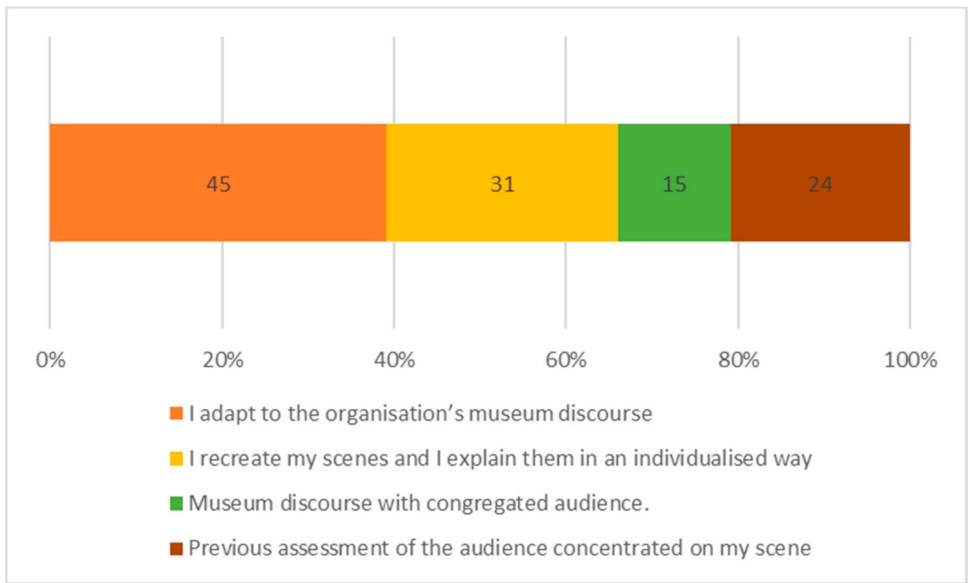

**Figure 3.** Model of public reenactment in the mediation.

Of the respondents, 46% believe the historical reenactment focuses on epic passages of the military past, compared to 22.6% who think they also highlight other types of elements (Figure 4). When the reenactors who have participated in the survey were asked about rigour in different aspects of the reenactment, they mention that military life and its material culture, with 86.9%, or the poliorcetic, with 66.1%, is developed or highly developed, compared to other facets such as diet (46.1%), music (29.6%) or the linguistic (19.1%) (Figure 5). The reenactors are capable of observing very clear differences between aspects that form part of the reenactment in relation to how rigorous they are. This demonstrates that the same level of care is not applied to all these aspects, those relating to war being the most meticulously cared for.

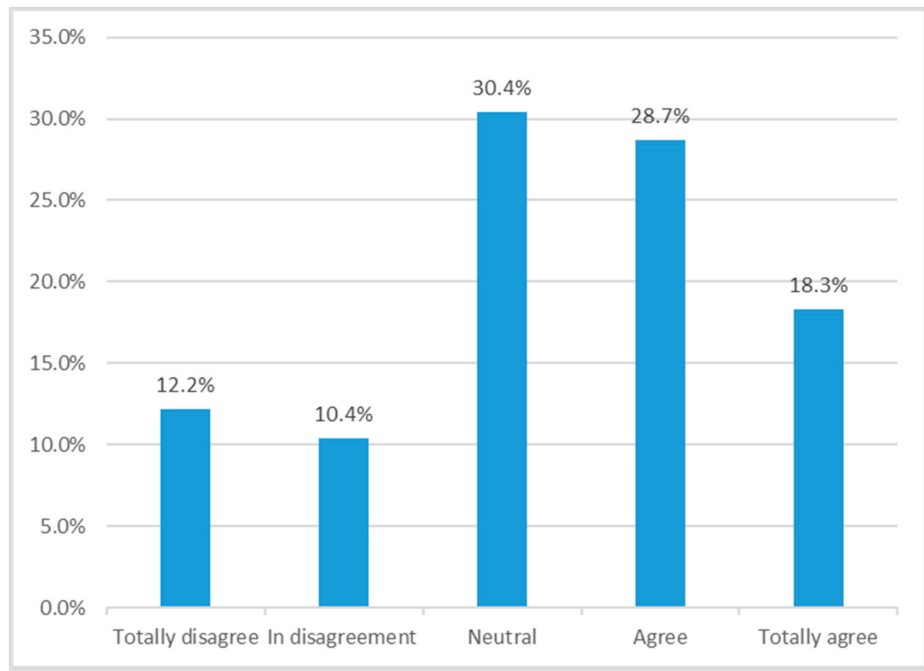

**Figure 4.** Historical reenactment focuses mainly on epic and glorious aspects and passages of the medieval military past.

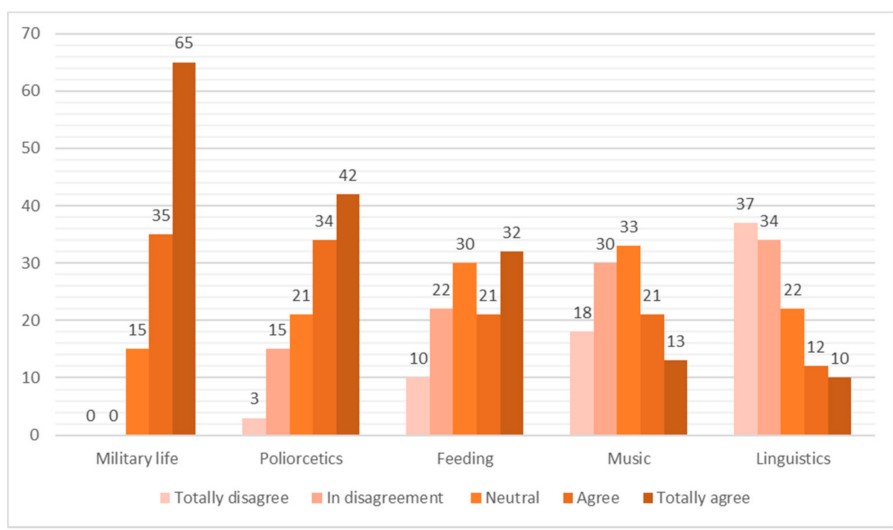

**Figure 5.** Rigour of the different aspects of the reenactment.

When asked the question, 'Is reenactment *per se*, without intermediation, capable of explaining complex historical structures?', the possible answers are very closely matched, but with 55.7% of negative answers. Nonetheless, 94.8% of these reenactors consider historical reenactment capable of complementing the teaching of more complex historical structures on war in the Middle Ages—such as structural causes and consequences, characteristics of a war-related phenomenon, military strategies, etc.—and is not exclusively limited to the cultural material it recreates. The answers acknowledge that reenactment alone cannot educate in said matters but, at the same time, indicate that it may have real educational possibilities, as long as it is used as a complementary tool in the educational process.

Within a historical reenactment event, the reenactors highlight interactivity as being fundamental (80%), they believe it awakens emotion in the spectator and the reenactor themself (94.8%) and allows them to understand the emotions and actions of the people from the past (80%). But when asked to choose which is most effective as an educational resource they opt for interactivity, with 61.7% preferring the reenactors to interact with the audience, either through dialogue and reflective processes—asking questions, conversing, allowing elements to be handled . . . —or performance processes. 25.2% prefer the use of empathy, attempting to get the audience to adopt the perspective of those who lived in a certain period in time. Finally, 13% prefer the use of emotions to explain and show elements of the past, causing the audience to engage with the educational process (Figure 6).

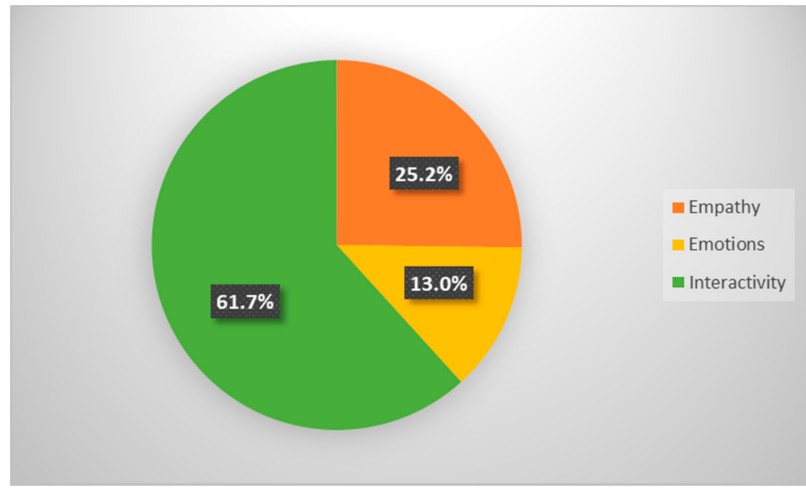

**Figure 6.** Most effective educational resources.

In relation to the teaching of medieval war through historical reenactment in heritage settings, we can state that 81.8% agree or completely agree with it taking place in the actual scenarios—battlefields, monuments, archaeological sites . . . —in which the history occurred. Precisely because of the fact that they took place in historical military sites and under the auspices of scientific methods, 60% of respondents state that historical reenactment must prioritise war education over other resources such as video games, literature or film or, at least be positioned on the same level (28.7%). Some even consider their practise a pastime without any investigative or educational faculties (11.3%) (Figure 7). Said results, in our opinion, reflect the stagnation between the professionalism/philanthropism that characterises historical reenactment. In a way, it is identifying the inability of a sector of the discipline to consider itself qualified to educate in the subject of war through its practise, whether conscious of the complexity and difficulty of the former, or not.

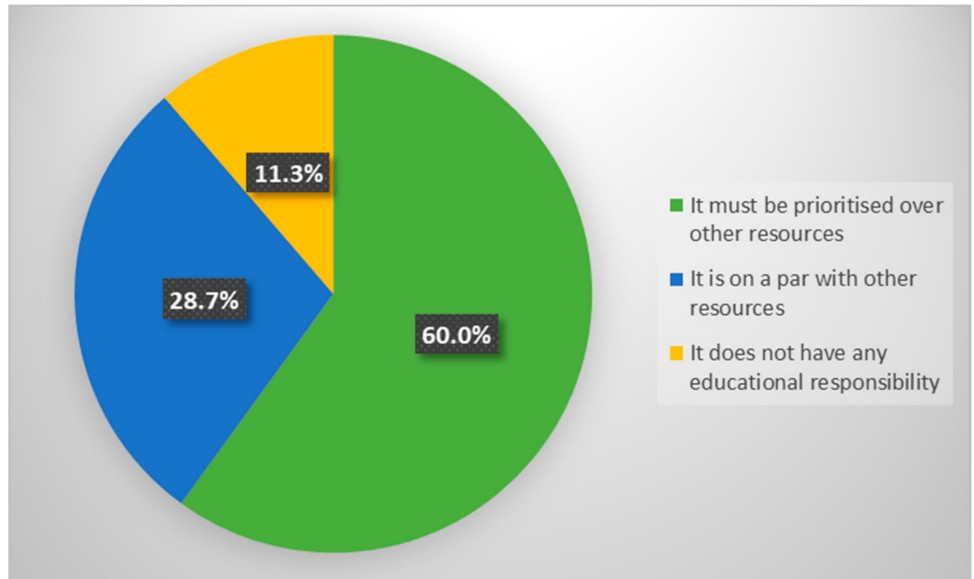

**Figure 7.** The role of historical reenactment in education on war.

From the current military science perspective, the war phenomenon must be considered from a triple dimension: strategic, tactical and technological. Thus, if we consider the study of war in the Middle Ages from this triple nature, the same scientific criteria should be applied to the approach used for its diffusion. Possibly linked to this increased historical rigour in creating the scene of the war and poliorcetic, as seen above, the reenactors believe that historical reenactment achieves a cultural approximation to the technology of war, teaching and showing replicas of the past (89.5%), but also a very accurate approximation to what military tactics were like in the Middle Ages (67.8%). However, do they achieve an approximation to what the complex military strategies were like in the Middle Ages, such as attrition warfare, siege warfare, geostrategy or military logistics? The majority believe so, with 50.5% agreeing or completely agreeing, but a significant group remains neutral (26.1%) or opposes this statement (23.5%). In relation to this question, the reenactors believe that medieval war may contribute to the development of competences such as strategic thinking. Based on the respondents' answers, we observed a broad consensus in considering reenactment an excellent tool to educate in technological material, obvious given that this discipline is mainly founded on the reconstruction of material culture. Nonetheless, a smaller percentage of the reenactors state that the actual practise itself is valid for teaching the tactical dimension, and an even smaller percentage sustain its validity for teaching the strategic dimension. The results (Figure 8) show that a large number of respondents is aware of the limitations of reenactment in this sense, but consider it optimal to educate in the subject of complex aspects. However, it is important to remember that it is not easy to reconstruct military tactics—on the battlefield, in poliorcetic spaces,

etc.—through reenactment; and it is practically impossible to organise learning contexts on military strategy—geostrategy, logistics, diplomacy, etc.—with material culture alone, without the support of more complex resources, which demonstrates: either a profound lack of knowledge about military science and therefore how to educate in medieval warfare, or that the concept the reenactor develops through their practise comes closer to dissemination—with the aid of a *cicerone* to explain these more intricate aspects- than to a complex museum and educational context.

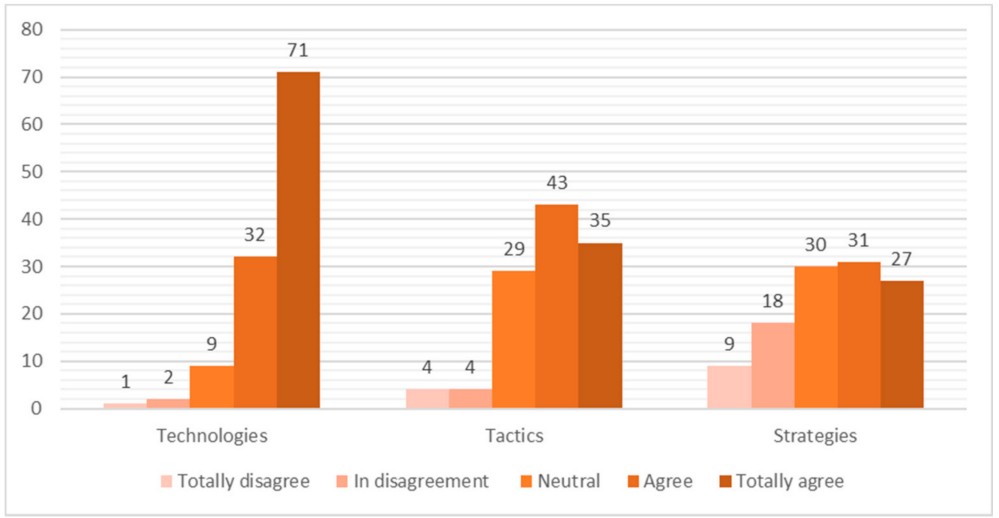

**Figure 8.** Historical reenactment achieves a very accurate approximation to what technologies, tactics and strategies were like.

These majority opinions are not reflected in other aspects such as education for peace. We should remember that in the theory arguments we indicated that education on war is largely avoided as it promotes violence. 90.3%, a wide majority of the reenactors believe that teaching war in the Middle Ages in particular, does not contribute to fostering violence and inciting the most execrable side of the human being. 47.9% agree or completely agree that historical reenactment educates for peace, insofar as it reflects on the suffering caused by war: the death, famine, disease, refugees, pain. On the contrary, 26% disagree or completely disagree with this statement. What does trigger more controversy is the question of how far to go with education for peace. Do we need to rebuild the discourse we have built on wars, avoiding teaching and delving into its most sordid side? This generates major bipolarisation of opinions, 34.8% against this question, versus a small majority, 39.1%, who would be willing to avoid this more sinister side. This data (Figure 9) shows us how the historical reenactors believe that war teaching does not foster violence, but at the same time they clearly believe it can be used to educate for peace, with doubts about whether or not to avoid the most miserable aspects in this teaching.

To conclude, another of the aspects worthy of note is that teaching about war in the Middle Ages may contribute to a comprehension of more complex historical structures, of which the war phenomenon is an essential piece (Figure 10). 94 out of 115 respondents either agree or completely agree with this statement, seeing the direct relationship between war and other aspects of the Middle Ages such as politics, economy or society.

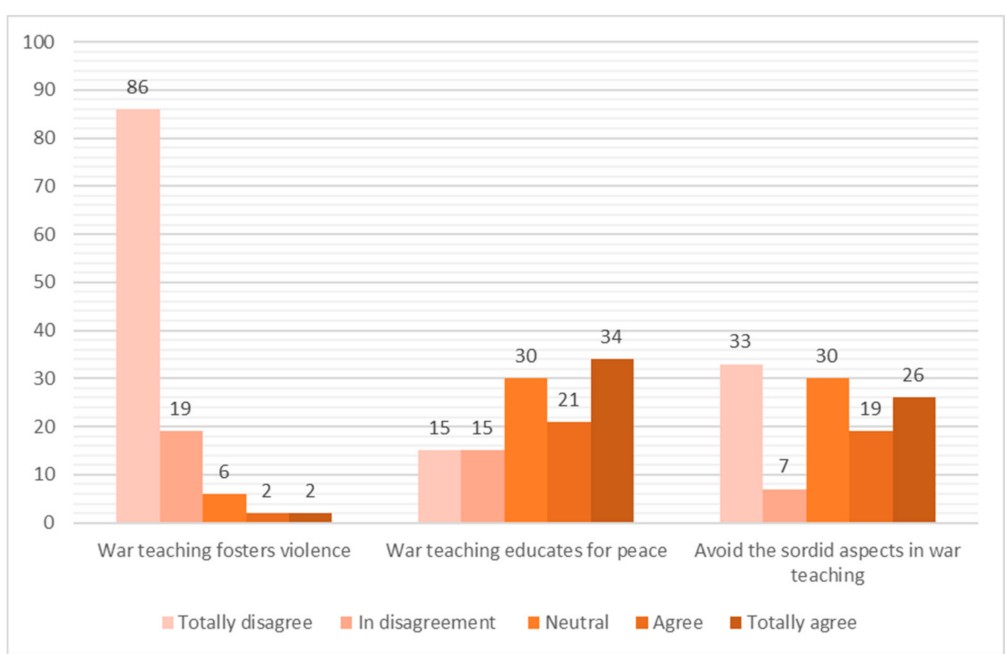

**Figure 9.** Education for peace.

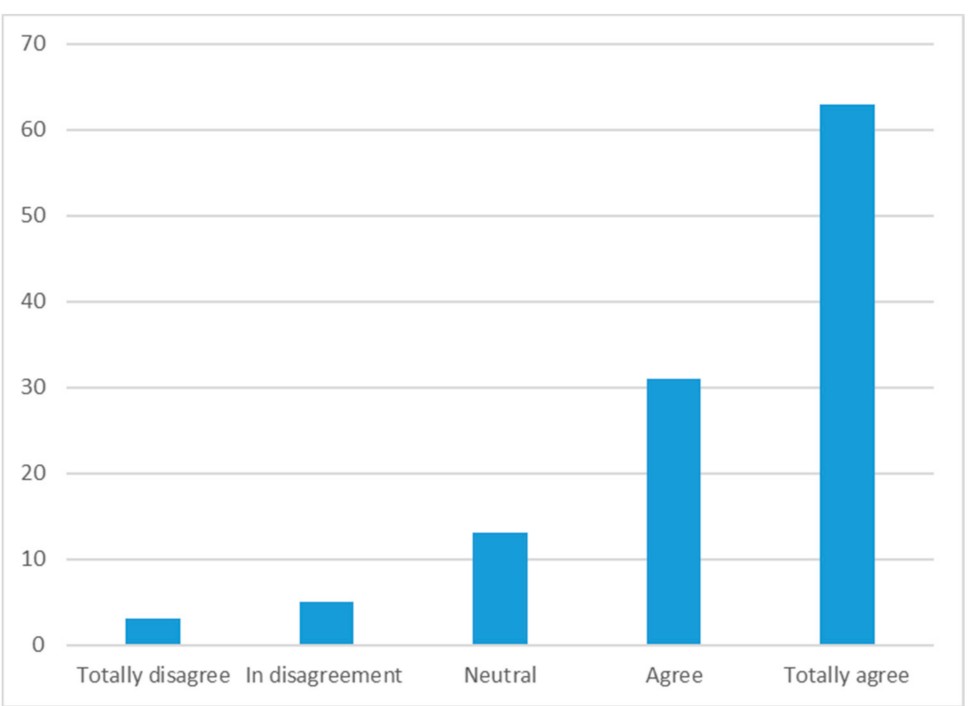

**Figure 10.** Comprehension of more complex historical structures through the teaching of war.

## 7. Discussion

We believe a reflection on the initial hypothesis of this research is an appropriate place to start the discussion of the study's findings, then subsequently analyse the secondary goals set down in the introduction section. The primary objective of our study consisted of an analysis of how the museum discourse of historical reenactment is structured when operating in military spaces of the Middle Ages, and the dissemination of the war phenomenon in that same period.

However, prior to this analysis, it is interesting to establish the objectives for which a historical reenactment activity is carried out and whether these are shared by the reenactors.

When asked about the objectives they deem appropriate for a historical reenactment activity, they mainly mention the socialisation of knowledge and the promotion of heritage as being fundamental. These objectives were already present in authors such as Cózar [78], Sebares [94], Español y Franco [95] and Rojas [77], among others. But in addition to these two objectives, others should also be mentioned including associative stimulation, the roots of identity [96] or constituting a research tool [79]. There are even objectives that come close to postures related to the so-called historical festivals and economic matters [97]. All these objectives appear among the reenactors' responses alongside others such as martial arts or the fun component and personal leisure.

Philanthropy is undoubtedly an element that conditions the performance of this practise, beyond the individual conception each exponent has of it. In fact, the double artist that historical reenactment presents: on the one hand, the heuristic to document reconstructed culture, and on the other the educational vocation, is affected by the dedication of its exponents, who may see it as a pastime that does not generally require specialisation from them. Nonetheless, the reenactors state they take their practise professionally in both dimensions. The truth is that as this training is not mandatory, aspects such as didactic design or educational planning in general may not meet the necessary requirements, also in the area of the war phenomena diffusion, obviously. This becomes apparent when many reenactors state they adapt to the museum requirements of those who manage the heritage sites they operate in, although at the same time they say they are the ones who draw up the discourse and the educational-communication strategies.

What does this data tell us? We observed that the exponents of historical reenactment make a huge effort to make the discipline a scientific praxis, in both of its dimensions: documentation and education. However, the lack of training in both areas and the fact that this is a philanthropic practise for them represents an obstacle to a certain degree for its consolidation as a top-tier tool for education and heritage management. We also believe it worth noting here that the separation of the discipline in Southern Europe from the institutions and centres of knowledge constitutes an added problem [79,81,82], which serves no other purpose than to worsen the existing situation.

It is important to remember that within this primary objective, we were also interested in a more in-depth exploration of which specific discursive aspects historical reenactment focuses on in its educational activity in relation to medieval war in heritage settings. To do so, the questionnaire probes how the historical period is managed within said educational processes, and whether the historical reenactment is capable of overcoming the finitude of its material nature. Thus, the reenactor shows an awareness that reenactment, *per se*, lacks the didactic resources that allow the teaching of historical time structures, and even complex historical processes inherent to shorter time periods. Nonetheless, they show an awareness of their own potential in this respect, and of the fact that the trend should be to veer towards these precepts.

More specifically, in relation to the field of heritage education, we observed a very interesting dichotomy. Again, it derives from the reenactors' own considerations on the educational fact they protagonise. For many, their action is exclusively limited to the area of dissemination, already noted by some authors at the time [78]. We observed this when a percentage of the respondents appears to want to maintain the academicism of the historiographic science in the discipline of historical reenactment itself. This is observed in the preference for a dialogue-based interactivity with the audiences, over and above other educational resources such as the use of emotions or empathy/perspective. The almost total preference for interactivity, on the other hand, as an educational vessel of primary importance, reveals that many reenactors are unaware of the use and application of other educational resources in the war education processes they carry out in the heritage site. Does historical reenactment not educate? Does it only provide data and knowledge without concerning itself with didactic transposition? We are undoubtedly looking at a consideration that may give rise to future lines of research.

If we focus on the analyses of the informal teaching of war in heritage sites of the Middle Ages, the answers also raise important debates. First and foremost, it catalyses the reenactors' own conception of the discipline they practise, when they consider that it should take precedence over other informal resources such as video games, film or television in the subject of warfare. The average reenactor here extols their role as exponents of a discipline that emerges as an appendix of the scientific contexts to vindicate themselves over other educational stimuli spread throughout our societies.

It is nonetheless necessary to bear in mind that in order to structure the educational planning of the military past of the Middle Ages, the three dimensions of military science we alluded to in the previous section must provide the backbone. Otherwise, we will be providing out of context information about the war phenomenon in general, removed from the postulates of military science. The answers show an obvious increase in the educational capacities inherent to reenactment as we move through the three dimensions, in which education in technological material is the best integrated in their own recreative praxis. As we noted above, reenactment is notoriously a discipline rooted in the reconstruction of material culture, meaning it is obvious that artefacts, objects and movable property heritage can be disseminated through this tool. As we probed deeper into the tactical and strategic dimension implicit in the war phenomenon, the reenactors acknowledged that dissemination of the same is more problematic. It is hard to understand how natural reenactments of people, artefacts and skills, on a 1:1 scale, can educate in the subject of tactical movements or military geostrategy, unless they use laborious displays that, ultimately, need to mould to fit the contemporaneity of our time, meaning the violent, ideological and cultural reality of a past that no longer exists is also blurred. Indeed, this is the same as the issue dealt with above, referring to whether this discipline is capable of overcoming its determinants to be able to teach complex aspects of the past. The subject of discussion worthy of particular note is, similarly, the fact that a large part of the reenactors interviewed accept that the reenactment *per se* is capable of tackling these more intricate issues, giving rise to the possibility of future lines of study to evaluate specific educational methods through tools such as direct observation, case studies, meta-analysis or audience studies. It is, in reality, the way to ascertain the extent to which historical reenactment takes on functions of dissemination peppered with the reproduction of material culture, or whether it actually has educational components allowing for structural components to be explored in depth.

With regards to the treatment of values and contents—many of a delicate nature-implicit in education in the subject of medieval war, the answers of the respondents tend to be more uniform. A higher proportion of reenactors admit to recreating aspects of the military past associated with "glorious" moments, even occasionally with identity-building or national historical passages. There is a general consensus that teaching war educates for peace. This is a consideration that we, as researchers, agree with. The concept of dissemination that the reenactor implements also comes close to a meticulous elaboration of the discourse, which we believe should be fostered above other informal stimuli that do not aim to disseminate or educate. Nonetheless, regarding how to approach what is taught about this war, the divided opinions become apparent. A small majority does not teach, argument or put forward sordid or gruesome aspects of the war-related past as they do not consider it necessary for the ultimate purpose of teaching the war phenomenon in its complexity.

## 8. Conclusions

In light of the data, certain conclusions linking the results obtained with the prospect of future research and even action should be included. We observed that the actual discipline, profiling itself as a very positive tool in the subject of heritage education, and specifically in relation to war in the Middle Ages, *per se* poses certain obstacles that must be considered. These hurdles—the philanthropic practise or its material nature- may affect the educational budgets and intervention we propose in war heritage, which occasionally implies the need

to provide specialised support to the practise. This support also becomes necessary when we wish to overcome the finitude of the mere reconstruction of material culture if we wish to teach structures of the historical time or complex aspects; it is, without a doubt, the unresolved issue this discipline must tackle.

It is, nonetheless, worth drawing attention to more specific aspects of the function of historical reenactment in heritage settings, begging the question of whether it should be considered an educational tool, in all of its aspects, or simply one more resource for dissemination, akin to the printed materials of a heritage site or its signage. This debate should take place henceforth, placing the emphasis on the actual exponents of the discipline, without forgetting that audience studies could also help evaluate the education processes that reenactment sets in motion.

To conclude, we believe it is important to put forward the prospect of a research study in relation to the military fact of the Medieval period in poliorcetic spaces, in light of the findings obtained. We understand that historical reenactment emerges as a resource worthy of consideration, as long as any museum project or intervention implementing it—or informal educational action—is aware of the limitations and strengths this study has enabled us to ascertain. These include the need to provide it with complementary resources for it to be effective in the teaching of complex historical processes; resources, which, in our opinion, should go beyond the use of a *cicerone* to translate what has been recreated, to encompass others of a graphic, technological or audio-visual nature, even though this may distort the nature and essence of the discipline in the eyes of the exponents and the public. Thus, as far as we are concerned, a whole range of actions allowing for the possibility of integrating historical reenactment into other educational and museum formats opens up to us. This is a prospect, therefore, that integrates the necessary and didactic discourse that accompanies the live reconstruction of history, with other elements that multiply its educational possibilities without compromising the traits that render it valuable, including interactivity, empathy or the use of emotions, among others.

**Author Contributions:** Conceptualization, D.E.-S. and J.G.F.-C.; methodology, D.E.-S. and J.G.F.-C.; investigation, D.E.-S. and J.G.F.-C.; writing—original draft, D.E.-S. and J.G.F.-C.; writing—review and editing, D.E.-S. and J.G.F.-C. All authors have read and agreed to the published version of the manuscript.

**Funding:** This research received external funding from the Grupo de Investigación ARGOS and the Instituto de Investigación en Ciencias Ambientales de Aragón. Both are from the University of Zaragoza.

**Data Availability Statement:** The data presented in this study are available on request from the corresponding author.

**Conflicts of Interest:** The authors declare no conflict of interest.

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
