# Peer review of "Education and Heritage of Medieval Warfare. A Study on the Transmission of Knowledge by Informal Educators in Defensive Spaces"

_education, doi:10.3390/educsci11070320_

Round 1

Reviewer 1 Report

It is a good effort and a very original research that links academical investigations to reenactment and public history. Two fields that, hopefully, will be developed in the future in history education:

Some comments to improve the paper:

  • In the 2nd epigraph of the theoretical frame: when teaching history and education for peace, the discussion is not just about showing (or not) wars and battles (and their motivations, causes, consequences, etc.) in the classroom, but to decide, as teachers, what kind of content we teach when speaking about wars and battles. To teach war is something compulsory in the curriculum  (and they appear in all text books), so the discussion could be focused in how to use that wars and introduce social contents to educate for peace. This is something discussed in Moreno-Vera (2020) "Historia pública y pensamiento histórico. Nuevos enfoques metodológicos para aprender la guerra de independencia española", and maybe this approach could be mentioned in the paper.
  • Regarding to the methods, It is important to deeply explain the number of reenactors participants that answered the survey
  • It is important to show the concrete questions included in the survey
  • It could be very desiderable to explain the validation of the survey. If it was validated by "group of experts" we need to know de comments and improvements they propose.
  • What was the process to analyze data? Do the authors use any kind of qualitative program to codificate the items? Atlas.TI, Aquad or any similar qualitative data package? Or did they use google forms?

Author Response

Dear colleague:

We are very grateful for your words and for the proposals to improve our article. We have added and improved the text with your instructions.

Thus, we have improved the investigation with the point of view of Moreno-Vera, which is very interesting. As you have indicated, it allows us to focus on how we use wars for an education for peace. A topic that, by the way, could serve to another interesting investigation and that we will surely address soon in our studies on medieval warfare and education.

On the other hand, we also find it very interesting to associate the project directed by Professor Rafael Zurita (Alicante University), in which Moreno-Vera's paper is framed, with which we have collaborated and we have an excellent academic relationship.

We have also added the rest of the questions that you propose, such as references to the group of experts that have validated the questionnaire, alluding to our research group. And in the same way we have added the qualitative program with which we have graphically materialized the answers, and the questions provided as an annex at the end of the study.

Thank you very much for your review and proposal to improve.

Greetings.

Reviewer 2 Report

I would reccomend to elaborate further on the role of associations and reenactments groups (p. 13), wehat are their challenges and struggles with educational activities. Perhaps some of the reenactments receive subsities form local /national authorities. Perhaps in such proposals there is an obligatory educational component (lecture, presentation, workshop).

Secondly, the report should  provide more information about the cooperation between museums and reeanactors, is this formal of informal (p. 14),  is the reenactment tailored made to the audience profile, what are the challenges that appearde due to the COVID -19 (online reenactment).

Author Response

Dear colleague:

We are very grateful for your words and for the proposals to improve our article. We have added and improved the text with your instructions.

Actually, your proposal about the “contractual” relationship that these groups have with local /national authorities in Spain or with the museum institutions plays a very important role in the educational processes that they implement in heritage, so we have included some reflections on this question.Similarly, we have also added information on recreation in times of COVID, based on information from the surveyed reenactors, to improve the paper.

Thank you very much for your review and proposal to improve.

Greetings.

Reviewer 3 Report

Education 1263618

Review of ‘Education and heritage of medieval warfare’

Thank you for the opportunity to review the manuscript. I thoroughly enjoyed reading it. This is a very well written (some English infelicities notwithstanding) and well-argued paper that advances the literature.

I have to offer only  minor, niggardly sounding comments:

In the Methodology section, I have been looking in vain for a comment on the approval of the survey by an institutional/university ethics board. There is also not such statement at the end of the paper. Given that the survey covers research of the opinions of human beings, such ethics approval is required for publication in journals.

In their discussion of the role of archaeology, and GIS, the authors may wish to consider the paper that advances such analytical concepts

Spennemann, D. H. R. (2020). Using KOCOA Military Terrain Analysis for the Assessment of Twentieth Century Battlefield Landscapes. Heritage, 3, 753-781

Of interest may also be an example of disarmament heritage as a counter-point:

Spennemann, D. H. R. (2021). Managing the Heritage of Arms Limitation Treaties. International Journal of Historical Archaeology. https://doi.org/10.1007/s10761-020-00584-2

Line 526:           What are “philanthropic criteria.” That makes little sense. I presume it is based on awkward translation from the Spanish into English.. philanthropic refers to organisations, not criteria. This needs to be addressed throughout

 of the

MINOR ISSUES

The manuscript needs a thorough edit by a native English speaker. There are some infelicities, e.g.

Line 112-113  ‘The answer is a rotund yes’  robust yes

Line 121            into goodies and baddies  put in quotation marks

Line 277 “The interfluves of the violent encounters” interfluves?

Author Response

Dear colleague:

We are very grateful for your words and for the proposals to improve our article. We have added and improved the text with your instructions.

We have added to the text that the questionnaire, although it is not a clinical study, it’s done accordance to the criteria, general principles and requirements of the Declaration of Helsinki on ethical principles for medical research in humans.

We have also added reflections about the Spennemann studies, as you have advised us. Thank you very much for these observations, because it is a topic that we consider very important that will surely occupy our lines of research in the future. 

Regarding the words you have observed, it is already in the hands of our translator (that we think is native…), with a general review. Thank you again for the guidelines.

Thank you very much for your review and proposal to improve.

Greetings.